# HTRA2/OMI-Mediated Mitochondrial Quality Control Alters Macrophage Polarization Affecting Systemic Chronic Inflammation

**DOI:** 10.3390/ijms25031577

**Published:** 2024-01-27

**Authors:** Qingqing Liu, Xiaoyu Yan, Yuan Yuan, Runyuan Li, Yuanxin Zhao, Jiaying Fu, Jian Wang, Jing Su

**Affiliations:** Key Laboratory of Pathobiology, Department of Pathophysiology, Ministry of Education, College of Basical Medical Sciences, Jilin University, 126 Xinmin Street, Changchun 130012, China

**Keywords:** HtrA2/Omi, macrophages, mitochondria, SCI, inflammation

## Abstract

Systemic chronic inflammation (SCI) due to intrinsic immune over-activation is an important factor in the development of many noninfectious chronic diseases, such as neurodegenerative diseases and diabetes mellitus. Among these immune responses, macrophages are extensively involved in the regulation of inflammatory responses by virtue of their polarization plasticity; thus, dysregulation of macrophage polarization direction is one of the potential causes of the generation and maintenance of SCI. High-temperature demand protein A2 (HtrA2/Omi) is an important regulator of mitochondrial quality control, not only participating in the degradation of mis-accumulated proteins in the mitochondrial unfolded protein response (UPRmt) to maintain normal mitochondrial function through its enzymatic activity, but also participating in the regulation of mitochondrial dynamics-related protein interactions to maintain mitochondrial morphology. Recent studies have also reported the involvement of HtrA2/Omi as a novel inflammatory mediator in the regulation of the inflammatory response. HtrA2/Omi regulates the inflammatory response in BMDM by controlling TRAF2 stabilization in a collagen-induced arthritis mouse model; the lack of HtrA2 ameliorates pro-inflammatory cytokine expression in macrophages. In this review, we summarize the mechanisms by which HtrA2/Omi proteins are involved in macrophage polarization remodeling by influencing macrophage energy metabolism reprogramming through the regulation of inflammatory signaling pathways and mitochondrial quality control, elucidating the roles played by HtrA2/Omi proteins in inflammatory responses. In conclusion, interfering with HtrA2/Omi may become an important entry point for regulating macrophage polarization, providing new research space for developing HtrA2/Omi-based therapies for SCI.

## 1. Introduction

SCI is the cause of noninfectious disease, a state of low-grade, persistent, noninfectious inflammation [1], involving autoimmune disease as well as neuroinflammation. In the absence of acute infectious injury or activation of pathogen-associated molecular patterns (PAMPs), SCI is usually triggered by damage-associated molecular patterns (DAMPs) [2,3,4]. Macrophages are important immune cells of the organism and are deeply involved in the regulatory process of SCI by virtue of their extensive plasticity, which is characterized by a complex balance between pro-inflammatory and anti-inflammatory responses [2]. Macrophages are classified into pro-inflammatory macrophages and anti-inflammatory macrophages, where pro-inflammatory macrophages are mainly involved in the maintenance of the pro-inflammatory state in the inflammatory response, while anti-inflammatory macrophages prevent excessive activation of the inflammatory response primarily through an anti-inflammatory effects [5]. In addition to their well-characterized gene and protein expression profiles, the metabolic processes of pro-inflammatory and anti-inflammatory macrophages are distinctly different. In the resting state, macrophages meet their energy requirements by oxidative phosphorylation (OXPHOS). After lipopolysaccharide stimulation, macrophages polarize toward pro-inflammatory macrophages, glucose uptake increases, and aerobic glycolysis becomes the major metabolic pathway [6]. In contrast to inflammatory macrophages that rely on glycolysis, IL-4 induces macrophage polarization into “anti-inflammatory macrophages with enhanced mitochondrial oxidative respiration”. Following IL-4 stimulation, anti-inflammatory macrophages are supplied with energy via fatty acid (FA) oxidation in addition to glucose consumption to support mitochondrial OXPHOS [7]. 

Mitochondria are key regulators of the cell fate during disease. UPRmt is a major pathway in cells that protects mitochondria and maintains homeostasis and normal function within the mitochondria. This pathway protects and repairs damaged mitochondria by transmitting signals of mitochondrial damage to the nucleus, which promotes the high expression of nuclear-encoded mitochondrial stress genes [8]. Mitochondrial fission and fusion ensure that damaged mitochondria are separated and removed in a timely manner and maintain the homeostasis of mitochondria and their contents [9]. Current studies have shown that mitochondria first initiate the fusion process to restore normal mitochondrial function; when mitochondrial fusion fails to maintain normal physiological function when the injury is further aggravated then mitochondrial fission is initiated to remove severely damaged mitochondria [10]. Studies have shown that many immune cells, including macrophages, alter their metabolism during inflammatory responses, and mitochondrial quality control (MQC) may be involved in these changes, leading to their close association with inflammatory diseases [11].

HtrA2/Omi is located in the mitochondrial membrane gap and was originally described as a serine protease that induces stress responses in mammalian cells [12]. Previous studies have mostly focused on the potential regulation of apoptosis by HtrA2/Omi proteins in neurodegenerative diseases [13], sarcopenia [14], and tumors [15]. With the progress of research, another role of HtrA2/Omi proteins in cells has been gradually discovered, as a key regulator of mitochondrial molecular quality control [16]. Studies have shown that HtrA2/Omi proteins can participate in the degradation of error-accumulated proteins in the UPRmt to maintain the normal mitochondrial function through the enzymatic activity they possess; they can also participate in mitochondrial kinetic-related protein interactions, maintenance of mitochondrial morphology, and many other intra-mitochondrial signaling cascades, suggesting that HtrA2/Omi proteins play a role in MQC. Rheumatoid arthritis (RA) is a systemic autoimmune disease involving SCI. In a collagen-induced arthritis mouse model, HtrA2/Omi modulated inflammatory responses in BMDM by enhancing the stability of TNF receptor-associated factor 2 [17]. In addition, inhibition of HtrA2 attenuated the lipopolysaccharide-induced inflammatory response and apoptosis in acute pneumonia in rats [18], suggesting that HtrA2/Omi changes act as inflammatory signaling factors involved in the SCI response process. Therefore, ameliorating the inflammatory environment by regulating the activation status of macrophages is an effective way to treat the disease.

In this review, we summarize the progress of studies by summarizing the role of HtrA2 in mitochondrial quality control and the imbalance of mitochondrial quality control to promote pro-inflammatory macrophages polarization. A possible mechanism by which HtrA2 deficiency promotes pro-inflammatory macrophages polarization is proposed, suggesting that targeting HtrA2 may be an important direction for the treatment of SCI.

## 2. Dysregulated Macrophage Polarization Promotes SCI Progression

The current study found that alterations in mitochondrial metabolism and physiology lead to macrophage activation into different states, such as alterations in oxidative metabolism, mtROS, mitochondrial ultrastructure, and membrane potentials [19]. With the release of ROS and damage-associated molecular patterns of mitochondrial origin, mitochondrial dysfunction initiates an inflammatory response by regulating the secretion of proinflammatory cytokines that interact with receptors, such as those involved in pathogen-related responses [20]. Thus, mitochondrial quality control plays an important role in the homeostasis of macrophage polarization and function by regulating mitochondrial morphostructure, function, lifespan, and bioenergetic properties. Macrophages act as key mediators of the inflammatory response and they are involved in the progression of several SCI diseases. Rheumatoid arthritis (RA) is a chronic disease that causes systemic inflammation, characterized by joint inflammation and damage; it is usually associated with an imbalance of pro-inflammatory and anti-inflammatory macrophages [21]. Elevated levels of anti-inflammatory macrophages have been associated with therapeutic response in RA and are considered an important therapeutic target for it [22]. Activation of inflammatory vesicles in microglia in the brain induces a shift in the metabolic pattern of microglia from oxidative phosphorylation to aerobic glycolysis and leads to polarization of microglia to a pro-inflammatory phenotype, which ultimately induces neuroinflammation and neurodegeneration and promotes the progression of Parkinson’s disease [23]. This suggests that abnormal macrophage polarization is an important factor influencing SCI. Coordination of the balance between pro-inflammatory and anti-inflammatory responses is essential for disease development, and dysregulation of macrophage polarization plays an important role in the development of excessive inflammation. Therefore, exploring the factors affecting macrophage polarization becomes a potential target for the treatment of inflammatory diseases.

## 3. HtrA2 Is a Novel Inflammatory Mediator and a Potential Target for Anti-Inflammatory Therapies

### 3.1. Structural Features of HtrA2

The highly conserved high temperature requirement A (HtrA) family of serine proteases has many different physiological functions and constitutes the core group of cellular proteases [24]. All four human HtrAs contain the characteristic motif of HtrA, a conserved protease domain and a C-terminal PDZ domain [25]. However, these enzymes show significant differences in their N-terminal regions, which may be necessary to accommodate their different functional properties. The families are naturally divided into two distinct groups based on their different N-terminal regions. The N-terminal regions of HtrA1, HtrA3, and HtrA4 contain a signal peptide (SP), an insulin growth factor binding protein domain (IGFBP), and a Kazal-type S protease inhibitor domain (Kaz) [25]. The N-terminal region of HtrA2 is completely different and includes a transient peptide (TP) and a transmembrane domain (TM) (Figure 1) [26]. 

Among the four identified human HtrAs, HtrA2 has been extensively studied for its enigmatic structural features and profound functional relevance. Studies have shown that HtrA2/Omi plays a role in the progression of neurodegenerative diseases, prostate cancer, and hepatocellular carcinoma [15,27]. Notably, HtrA2/Omi is involved in the development of several SCIs including rheumatoid arthritis (RA) and neuroinflammation. Jeong et al. showed that during endoplasmic reticulum stress-induced apoptosis, synoviocytes release HtrA2 into the synovial lumen of the RA, which is a key regulator of the production of inflammatory cytokines [28]. HtrA2 contributes to the pathogenesis of autoimmune arthritis through the differentiation of Th17 cells by STAT3 [29]. Motor neuron degeneration (mnd)2 mice lack HtrA2 activity due to a missense mutation Ser276Cys in the structural domain of the HtrA2 protease [30]. Lee et al. found that HtrA2^mnd2^ mice significantly increased pSTAT3 expression, and HtrA2 induced STAT3 degradation in vitro, demonstrating that HtrA2 improves RA by inhibiting STAT3 [31]. Hu et al. showed that increased transcription of pro-inflammatory genes, as well as activation of astrocytes and microglia, were observed in the brain of HtrA2^mnd2^ mice, suggesting that HtrA2/Omi is an intrinsic cytokine that inhibits neuroinflammation [32]. In addition, HtrA2/Omi expression correlates with the level of infiltration of multiple immune cell populations [33]. However, there is currently little understanding of the molecular mechanisms by which HtrA2/Omi controls inflammation and immune responses are unknown. Studies have shown that HtrA2/Omi plays a role in mitochondrial homeostasis [34]. Thus, the molecular mechanisms by which HtrA2/Omi controls inflammation and immune responses may be closely related to the role of HtrA2/Omi in mitochondrial homeostasis.

### 3.2. Functions of HtrA2 in MQC

Mitochondria are the primary site of cellular energy supply, where most cellular ATP is produced by oxidative phosphorylation in mitochondria, accounting for more than 80% of the energy required for life activities. Also, mitochondria are key regulators of cell fate during disease. They control cell survival by producing ATP that promotes cellular processes and, conversely, control cell death by inducing apoptosis through the release of pro-apoptotic factors (e.g., cytochrome C). Therefore, stringent quality control mechanisms must be in place to ensure a healthy mitochondrial network. Current studies have identified a central role for mitochondria in SCI aseptic inflammation [35]. MQC plays a key role in these processes. In the MQC system, there exists an important proteasome, HtrA2/Omi, located in the mitochondrial membrane interstitial space, originally described as a serine protease that induces a stress response in mammalian cells [12] and coordinates the MQC system by degrading its specific substrates thereby [36]. The current study shows that HtrA2/Omi promotes cell survival by maintaining mitochondrial homeostasis [30,37,38].

#### 3.2.1. HtrA2 Dysregulation Triggers UPRmt

HtrA2 is a key stress-protective protease that cleaves misfolded proteins in an ATP-independent manner to protect cells from stress induced by toxic protein aggregates [39]. When proteins do not fold correctly, intermediate ‘misfolded’ forms called unfolded protein deposits (UPODs), stick together to form protein aggregates. There is growing evidence that HtrA2 exerts prominent neuroprotective effects by degrading toxic aggregates in mitochondria and that its aberrant function leads to mitochondrial dysregulation [40,41,42,43]. HtrA2/Omi is involved in the degradation of misfolded mitochondrial proteins, thus contributing to the maintenance of cellular homeostasis [36]. Felicity et al. showed that in neuronal cells HtrA2 deficiency has been shown to result in the accumulation of unfolded proteins in the mitochondria, a decrease in mitochondrial respiration, an increase in ROS production, and ultimately neuronal cell death [44]. Recent studies on isolated mitochondria from HtrA2/Omi knockout mice have shown increased accumulation of unfolded subunits of respiratory complexes I-IV and systemic respiratory chain dysfunction [42]. In addition, HtrA2/Omi is a protein quality control protease important for mitochondrial homeostasis and plays a major role in protein quality control and maintenance of cellular homeostasis. It has been shown that one of the main triggers of UPRmt is a disturbance of protein stabilization in the mitochondrial intermembrane space (IMS) [41]. Luena et al. showed that accumulation of proteins in the mitochondrial intermembrane space in the breast cancer cell line MCF-7 activates a unique UPRmt [45]. Under physiological conditions, proteins accumulated in the IMS are degraded by IMS proteases such as HtrA2, OMA1, and Yme1L [41,46,47]. The above suggests that HtrA2/Omi is a protease that is localized to the IMS in the physiological state and is involved in MQC to maintain mitochondrial protein homeostasis [45,48]. HtrA2 deficiency increases protein accumulation in the IMS to promote UPRmt formation.

UCF 101 is a HtrA2-specific inhibitor [49]. Klupsch et al. found that in MEF cells, UCF 101 activated CHOP transcription and expression of UPR response-associated protein [50]. It can lead to accumulation of unfolded proteins in mitochondria, oxidative stress, and defective mitochondrial respiration [51]. In addition, Meng et al. showed increased expression of the UPRmt-related proteins HSP10, HSP60, and LONP in HtrA2Hetero mice [52], suggesting that HtrA2/Omi deletion leads to MQC imbalance through an increase in UPRmt. UPRmt activation of PKR (double-stranded-RNA-activated protein kinase) is dependent on ClpP activity; Pkr^−/−^ mice were unable to activate the UPRmt target gene CPN60 in intestinal epithelial cells after the administration of dextran sodium sulfate and were protected against DSS-induced colitis [53]. Induction of PKR links the mitochondrial unfolded protein response to the pathogenesis of intestinal inflammation. In addition, NF-κB is also upregulated when the UPR is activated [54]. Thus, the UPRmt may be involved in the balance between inflammatory and anti-inflammatory responses.

#### 3.2.2. HtrA2 Dysregulation Induces an Imbalance in Mitochondrial Dynamics

Mitochondria can form complex networks within cells. Depending on the health of the mitochondrial network and the energy requirements of the cell, the size, morphology, and number of mitochondria are constantly changing [55]. The number and morphology of mitochondria are controlled by tightly regulated fission and fusion events [56]. Mitochondrial fusion maintains mitochondrial function and prevents the production of defective mtDNA [57]. Fusion is mediated by fusion proteins in the mitochondrial outer and inner membranes (OMM and IMM), termed mitofusin 1 (Mfn1), mitofusin 2 (Mfn2), and optic atrophy 1 (OPA1), respectively [58]. Mitochondrial fusion is a two-step process that involves OMM fusion mediated by Mfn1/2 and inner mitochondrial membrane (IMM) fusion mediated by OPA1. Mfn1 and Mfn2 share significant sequence and structural similarities, but they appear to have distinct cellular functions [59]. Mfn1 more efficiently mediates mitochondrial fusion, which may be due to having approximately eight times the GTPase activity [60]. OPA1 also relies on Mfn1, but not Mfn2, to promote fusion [58]. The fusion of two mitochondria produces hyperfused mitochondria, which can be stimulated by abnormal stimuli to produce more ATP. Mitochondrial fission is controlled by dynamin-related protein 1 (DRP1), which interacts with mitochondrial fission 1 (FIS1) after recruitment to the OMM to promote mitochondrial contraction and eventual fission [61,62]. 

Kieper et al. found that HtrA2/Omi interacts directly with OPA1 both functionally and physically, and that OPA1 levels were significantly increased in XMEF of HtrA2/Omi knockout mice. Transfection of HtrA2/Omi into HtrA2/Omi knockout cells restored increased OPA1 levels to those in wild-type (WT) mice [63]. These results demonstrated that HtrA2/Omi deletion is involved in regulating mitochondrial kinetic processes through the promotion of mitochondrial fusion, which affects disease progression. In addition, it has been shown that silencing of Yes-associated protein drives JNK phosphorylation, which induces Drp1 activation and translocation to the mitochondrial surface, thereby initiating mitochondrial fission. Excessive mitochondrial fission mediates the leakage of HtrA2/Omi from the mitochondria into the cytoplasm, demonstrating that excessive mitochondrial fission promotes HtrA2/Omi mitochondrial leakage into the cytoplasm [64] which in turn produces HtrA2/Omi-reduced mitochondria, further exacerbates mitochondrial fusion, and promotes MQC imbalance (Figure 2). The above suggests that HtrA2/Omi is associated with mitochondrial fission, but whether HtrA2/Omi deficiency regulates the mitochondrial fission response requires further verification.

The current study suggests that mitochondrial dynamics-related MQC is associated with the activation of the inflammatory response [65]. An imbalance between mitochondrial fusion and fission leads to mitochondrial dysfunction, which in turn results in mitochondrial fragmentation, diminished oxidative phosphorylation energy supply, reduction of mtDNA, and increased ROS production. This ROS burst represents a major pro-inflammatory factor by regulating NF-κB expression and activity [66]. Abnormal mitochondrial fusion and division activate different early inflammatory mediators such as tumor necrosis factor α, interleukins, and IFN-γ [67]. Mitochondrial dysfunction with upregulation of mitochondrial fission prevents the elimination of damaged mitochondria; the generation of ROS in excess of antioxidant activity may be an initiating factor for SCI [68]. With the release of ROS and mitochondria-derived DAMPs, mitochondrial dysfunction initiates an inflammatory response by promoting the release of proinflammatory cytokines that interact with DAMPs receptors [20]. 

## 4. HtrA2-Mediated Modulation of Mitochondrial Quality Control Signaling Is Involved in the Regulation of Macrophage Polarization Phenotype in SCI

HtrA2/Omi activity is critical for the inflammatory response. Inflammation and oxidative stress interact with each other, with inflammation increasing ROS production and ROS exacerbating inflammation [69]. HtrA2/Omi is normally present in the mitochondrial membrane interstitial space; however, under stressful conditions such as oxidative or heat stress, it disrupts the inner membrane and migrates into the mitochondrial matrix [70]. This results in a decrease in HtrA2/Omi in the membrane interstitial space. Under apoptotic signaling, HtrA2/Omi is released from the mitochondria into the cytoplasmic lysate and is involved in disease progression in neurodegeneration and rheumatoid arthritis [27]. In addition, in endoplasmic reticulum stress-induced RA-FLS, HtrA2 is released from mitochondria to the extracellular space [28]. It suggests that long-term exposure of macrophages to oxidative stress and heat stress within the SCI response releases HtrA2/Omi from the mitochondrial membrane interstitial space to the mitochondrial matrix, cytoplasm, and extracellularly, which ultimately leads to a decrease in mitochondrial membrane interstitial HtrA2/Omi.

Data obtained from the database BioGPS suggest that HtrA2 levels are high in immune cells, especially precursor cells of myeloid-derived immune cells. Mitochondrial membrane gap HtrA2/Omi deficiency reduces pro-inflammatory cytokine production in LPS or CpG-triggered BMDM, suggesting that HtrA2/Omi is an immune response regulator [17]. Current studies have found that alterations in mitochondrial metabolism and physiology lead to macrophage activation into different states, such as alterations in oxidative metabolism, mtROS, mitochondrial ultrastructure, and membrane potential [19]. HtrA2/Omi plays an important role in the homeostasis of macrophage polarization and function by modulating the morphostructure, function, lifespan, and bioenergetic properties of mitochondria.

### 4.1. Aberrant HtrA2 Function Promotes ROS Generation Inducing Ox-mtDNA-Catalyzed Macrophage Immune Activation

Studies have shown that mitochondrial DNA (mtDNA) regulates energy production and cellular metabolism, and is a key signaling molecule that triggers the inflammatory response. mtDNA contains inflammatory unmethylated CpG motifs similar to those found in bacterial DNA, which makes it more of an “exogenous” than an “endogenous” DNA [71]. Without histone protection, mtDNA is more susceptible to mitochondrial reactive oxygen species (mtROS) attack [72], leading to oxidative damage to mtDNA. Min-Kyung et al. showed that HtrA2 regulates α-Synuclein-mediated mtROS production in microglial cell mitochondria. HtrA2 inactivation led to an increase in mtROS in microglia [73]. Restoration of HtrA2/Omi expression rescued CCl4-induced hepatic fibrosis in hepatocytes and reversed mitochondrial dysfunction (e.g., excessive ROS production) [51]. This suggests that HtrA2/Omi acts as an important proteasome in the MQC system to regulate mtROS production.

Increased mitochondrial ROS release inhibits macrophage anti-inflammatory function. ROS promote macrophages to undergo pro-inflammatory macrophages polarization. Mitochondria are the main source of ROS in macrophage polarization [74]. It has been shown that decreased mitochondrial ROS levels inhibit pro-inflammatory macrophages polarization and promote macrophage polarization to anti-inflammatory macrophages [75]. NADPH oxidase 4 (NOX4) plays an important role in the transport of electrons in the mitochondrial respiratory chain, which is one of the major sources of ROS. In a study of inflammatory bowel disease it was shown that NOX4 is able to promote polarization of intestinal macrophages in the pro-inflammatory macrophages direction through increased ROS production [76], and that elevated glucose uptake induces an overproduction of ROS, leading to macrophage activation in the pro-inflammatory direction in vitro [77,78]. Furthermore, systemic inflammation with predominant pro-inflammatory macrophages polarization was observed in Ndufs1 knockout (a key component of mitochondrial complex I) mice, which could be attenuated by the use of the mitochondria-targeted antioxidant Mito-TEMPO. This supports a critical role for mitochondrial ROS in macrophage polarization to pro-inflammatory macrophages [79]. Thus, ROS are critical for anti-inflammatory macrophages polarization, while mitochondria also play an important role in ROS. So, how does ROS regulate macrophage polarization?

Goo et al. showed that in MEF cells and mouse brain tissues, HtrA2 protein deletion led to mtDNA conformational changes by promoting mtROS production, targeting only HtrA2/Omi overexpression; mtDNA repair genes, including POLG2, Twinkle, and APTX1 were significantly up-regulated in HtrA2/Omi knockdown cells, demonstrating that HtrA2/Omi was able to restore mtDNA conformational stability in HtrA2/Omi knockdown MEF cells [34]. Hui-Gwan et al. showed that HTRA2 helps to maintain mitochondrial DNA conformation changes in MEF cells and HeLa cells, and HtrA2/Omi deficiency through ROS production and mutation leading to mtDNA damage [80]. We speculate that this mtDNA conformational change is associated with resistant oxidative-damaged mtDNA (Ox-mtDNA). It has been shown that HtrA2/Omi deficiency induces Ox-mtDNA production by promoting mtROS survival. Increased ROS cause an alteration in mitochondrial membrane potential, which subsequently induces the opening of the mitochondrial permeability transition pore (MPTP) [81], which releases Ox-mtDNA into the cytoplasm, where it is then recognized by the cytoplasmic receptor for DAMPs [82]. DAMPs activate pathways related to innate immunity through toll-like receptors (TLR) [83,84]. MtDNA contains an unmethylated CpG-DNA sequence; these sequences are TLR9 ligands that can mediate inflammation through the NF-κB pathway [85]. TLR9 is a classic TLR that recognizes CpG DNA and senses nucleic acids to initiate myeloid differentiation primary response protein 88 (MyD88) triggering a downstream signaling cascade that activates the nuclear transcription factor NF-κB [86], promotes the production of pro-inflammatory cytokines such as interleukin-1β (IL-1β) and interleukin-12 (IL-12), as well as interferon I (IFN), and promotes macrophage polarization toward pro-inflammatory macrophages (Figure 3) [87,88].

### 4.2. OXPHOS Complex Abnormalities Promote Pro-Inflammatory Macrophages Polarization

MQC has a profound effect on macrophage function. Mitochondria are the main site of cellular energy supply, in which most cellular ATP is produced by oxidative phosphorylation in the mitochondria, accounting for more than 80% of the energy required for life activities. In the resting state, macrophages meet their energy requirements through oxidative phosphorylation. After LPS stimulation, macrophages are polarized towards pro-inflammatory macrophages; the tricarboxylic acid cycle, fatty acid oxidation, and oxidative phosphorylation activity of the pro-inflammatory macrophages subpopulation is impaired and relies on increased levels of glycolysis (also known as the Warburg effect) and the pentose phosphate pathway to satisfy the cellular energy requirements [6]. Cytokines such as tumor necrosis factor α, IL-6, and IL-1β and chemicals such as ROS and NO are characteristic of pro-inflammatory macrophages [89].

One of the hallmark features of HtrA2 as a stress-protective protease is its ability to cleave aggregated proteins [42,90,91]. HtrA2/Omi deficiency promotes the generation of unfolded proteins, in particular components of respiratory complexes on mitochondrial membranes. The current study shows that HtrA2/Omi protease deficiency leads to oxidative phosphorylation and a reduction in the number of mitochondrial genes encoding ETC subunits [14]. Moisoi et al. showed that an increase in the unfolded subunits of the HtrA2/Omi KO brain mitochondrial complexes I-IV leads to systemic respiratory dysfunction [42]. This suggests that when the protease of HtrA2 activity is absent, the inability to remove unfolded respiratory complex proteins leads to mitochondrial complex abnormalities that inhibit OXPHOS levels. In addition, mitochondrial respiratory chain complexes I and II were significantly reduced in HtrA2^mnd2^ littermate mice, and mitochondrial synthesis of ATP energy was reduced, leading to increased ROS production, which further impaired the respiratory chain complexes and exacerbated the mitochondrial damage, resulting in a vicious cycle [92]. It was shown above that HtrA2/Omi deficiency leads to abnormal mitochondrial dynamics by promoting mitochondrial division. In the hippocampus of heart failure mice, mitochondrial fission gene DRP1 has been reported to be increased, and damaged mitochondria release excess ROS and have a reduced ability to produce ATP [93]. Furthermore, reduced mitochondrial mass and activity was found after overexpression of mitochondrial fission factor in the breast cancer cell line MCF7, and analysis of metabolic fluxes using Seahorse XFe96 revealed reduced levels of oxidative phosphorylation and glycolysis in this cell [94]. This suggests that increased mitochondrial fission due to HtrA2/Omi deficiency may impair mitochondrial energy metabolism and thus increase macrophage polarization towards pro-inflammatory macrophages.

In summary, HtrA2/Omi plays an important role in the regulation of macrophage activation by affecting MQC. MQC abnormalities affect the direction of macrophage polarization regulated by ROS generation on the one hand; on the other hand, they lead to abnormal energy metabolism driving the direction of pro-inflammatory macrophages polarization.

## 5. Conclusions

Systemic SCI is a low-grade, persistent inflammation that may cause tissue and organ damage through, for example, oxygen stress. The persistence of inflammatory factors (e.g., IL-1β, IL-6, and TNF-α) and damage to tissues is the underlying cause of SCI. MQC is an integrated network that monitors the quality of mitochondria and an endogenous cellular protective program that is essential for maintaining mitochondrial homeostasis and function. Many mitochondrial components and metabolites can function as DAMPs and promote inflammation upon release into the cytoplasm or extracellular environment. As mentioned previously, in SCI, oxidative stress and other induced release of mitochondrial membrane interstitial HtrA2/Omi into the mitochondrial matrix or cytoplasm leads to a reduction of HtrA2/Omi in the mitochondrial membrane interstitial space. One of the hallmark features of HtrA2 as a stress-protective protease is its ability to cleave aggregated proteins, which in turn promotes UPRmt disruption and a reduction in the OXPHOS complex, contributing to an increase in ROS; on the other hand, a decrease in HtrA2/Omi promotes an increase in OPA1, which in turn promotes mitochondrial fragmentation leading to mitochondrial kinetic derangement, contributing to increased ROS release. The increased ROS causes oxidative damage to mtDNA, and Ox-mtDNA escapes from the mitochondria to the cytoplasmic matrix or even extracellular space. Ox-mtDNA acts as DAMPs to initiate TLR9/MyD88/NF-κB, promoting pro-inflammatory macrophages polarization and releasing pro-inflammatory cytokines, which further amplifies the inflammatory response. Targeting HtrA2/Omi is therefore a new direction for the treatment of SCI.

The SCI response is associated with mitochondrial dysfunction as well as high levels of ROS production [95]. It has been shown that impaired mitochondrial quality control due to loss of function of the mitochondrial protease HtrA2 is associated with the accumulation of damaged mitochondria [96]. Neurodegenerative phenotypes in mice completely lacking HtrA2/Omi or expressing an enzymatically inactivated mnd2 mutant suggest that the proteolytic activity of HtrA2/Omi exerts a protective role in the mitochondria of neuronal cells [30,37]. Mitochondria are the primary site for the production of ROS or free radicals, which are necessary to fight infection. However excessive and uncontrolled ROS production can be detrimental to cells, leading to mitochondrial and tissue damage [95]. The results of the study showed that BMDMs with dysregulated mitochondrial quality control have high levels of ROS production, suggesting that high levels of oxidative stress mediated by dysregulated mitochondrial quality control may serve as an additional important pathway for the maintenance of SCI. In addition, the high levels of ROS produced could in turn act as new damage factors to make the already precarious mitochondrial quality control even more vulnerable to mitochondrial quality control dysregulation and oxidative damage promoting inflammatory progression. It has been shown that HtrA2, in addition to its classical role as a serine protease in mitochondria, can acquire new functions as a regulator of inflammation, demonstrating the functional diversity of mitochondrial proteins. Inhibitors of apoptosis (IAPs) proteins were initially characterized for their ability to directly bind and inhibit apoptotic caspases. Interestingly, the current study demonstrates the importance of IAPs in modulating inflammatory signaling and immune responses [97,98]. It was shown that inhibition of cIAPs modulated the activity of tumor-associated macrophages and reversed the status of anti-inflammatory macrophages to pro-inflammatory macrophages, with massive recruitment of neutrophils [99]. At the same time, levels of pro-inflammatory cytokines, including TNFα, IFNγ, and IL-12, were increased and promoted SCI progression. In addition, Kavanagh et al. suggested that cIAP-2 may be a therapeutic target to reduce the pro-inflammatory activation of microglial cells [100]. Many articles have shown that cytoplasmic HtrA2 can bind and antagonize IAPs [40,101]. Thus, HtrA2 may affect SCI by regulating proteins of the IAP family.

This review highlights the importance of HtrA2/Omi as an important regulator of mitochondrial quality control, in addition to its involvement in protein degradation by mis-accumulation through the UPRmt and in the regulation of mitochondrial dynamics. In addition, HtrA2/Omi is involved in the regulation of mitochondrial biogenesis and mitochondrial autophagy of HtrA2/Omi to renew the mitochondrial reservoir by interacting with the GSK3β/PGC-1α pathway. On the one hand, Xu et al. showed that Omi cleaves glycogen synthase kinase 3β (GSK3β), a kinase that promotes the degradation of PPARγ coactivator-1α (PGC-1α), and thus regulates PGC-1α, an important factor in mitochondrial biogenesis; moreover, in HTRA2^mnd2^ mice, the abundance of GSK3β was significantly increased, while the abundance of PGC-1α was significantly decreased [102]. On the other hand, Kang et al. showed that HTRA2^mnd2^ mice exhibited marked cardiac enlargement with left ventricular hypertrophy, which was associated with increased autophagosome activity in cardiac tissue. The increase in autophagosome activity may be due to an increase in mitochondrial autophagy, as their in vitro experiments showed that two mitochondrial proteins, PPID/peptidylprolyl isomerase D (cyclophilin D) and SOD2/MnSOD, were degraded at an increased rate in HTRA2-expressing MEFs in response to starvation compared to HTRA2-deficient mouse embryonic fibroblasts (MEFs) [103]. In addition, Cilenti et al. showed the accumulation of Mulan protein in various tissues of HTRA2^mnd2^ mice as well as in HTRA2-deficient MEFs, which resulted in a significant reduction of mitofusin 2 (Mfn2) protein and an increase in mitochondrial autophagy [104]. Furthermore, this work highlights the importance of HtrA2 as a potential therapeutic target for SCI, as it is a common trigger of the inflammatory process. HtrA2/Omi is a potential biomarker for SCI. However, its therapeutic utility is speculative and further studies are needed to fully grasp its role. The location of elevated HtrA2/Omi varies in different inflammatory diseases, with higher concentrations of HtrA2 in rheumatoid arthritis (RA) synovial fluid (SF) than in osteoarthritis (OA) SF, which correlates with the number of immune cells in RA SF, and with the finding that treatment of RA FLS with an endoplasmic reticulum stress inducer resulted in the release of HtrA2/Omi from the mitochondria into the extracellular space [28]. In addition, HtrA2/Omi was released from mitochondria to the cytoplasm in pneumonia [18]. Therefore, targeting drugs to the specific location of HtrA2/Omi expression in different inflammatory conditions is more therapeutically promising.

## Figures and Tables

**Figure 1 ijms-25-01577-f001:**
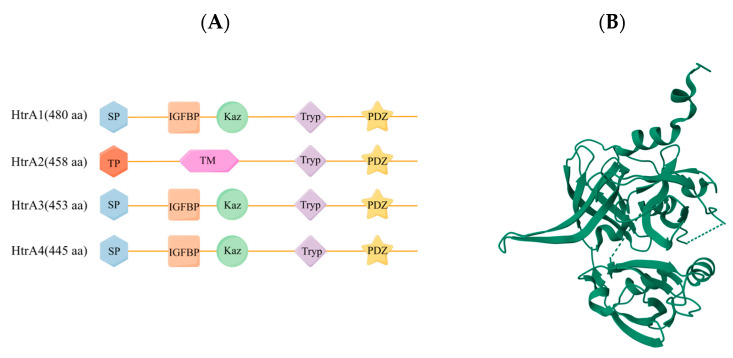
Schematic representation of the protein structural domain organization of the four human HtrAs. (**A**). Comparison of the structural domain organization of the human HtrA family, including HtrA1, HtrA2, HtrA3, and HtrA4. (**B**). HtrA2 crystal structure (PDB ID: 1LCY). By Figdraw (https://www.figdraw.com, accessed on 1 December 2023).

**Figure 2 ijms-25-01577-f002:**
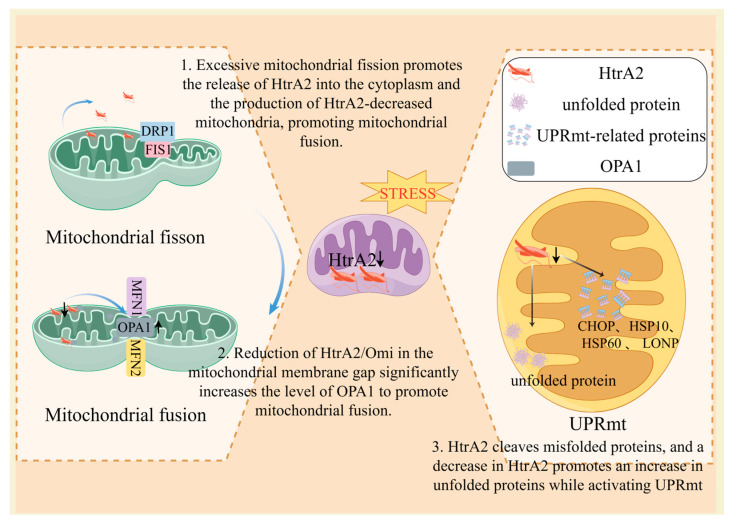
Functions of HtrA2 in MQC. Macrophages exposed to SCI for a long period of time release HtrA2/Omi from the mitochondrial membrane gap into the mitochondrial matrix, cytoplasm, and extracellularly under stress conditions such as oxidative stress and heat stress, which ultimately leads to a decrease in mitochondrial membrane gap HtrA2/Omi. HtrA2/Omi interacts directly with OPA1 functionally and physically, and HtrA2/Omi decreases OPA1 levels significantly and inhibits mitochondrial fusion. Thus, a decrease in HtrA2/Omi in the mitochondrial membrane gap would allow an increase in OPA1. Excessive mitochondrial division promotes the release of HtrA2 into the cytoplasm and the production of HtrA2-decreased mitochondria, promoting mitochondrial fusion. Furthermore, HtrA2 cleaves misfolded proteins, and a decrease in HtrA2 promotes an increase in unfolded proteins while activating UPRmt. By Figdraw (https://www.figdraw.com, accessed on 1 December 2023).

**Figure 3 ijms-25-01577-f003:**
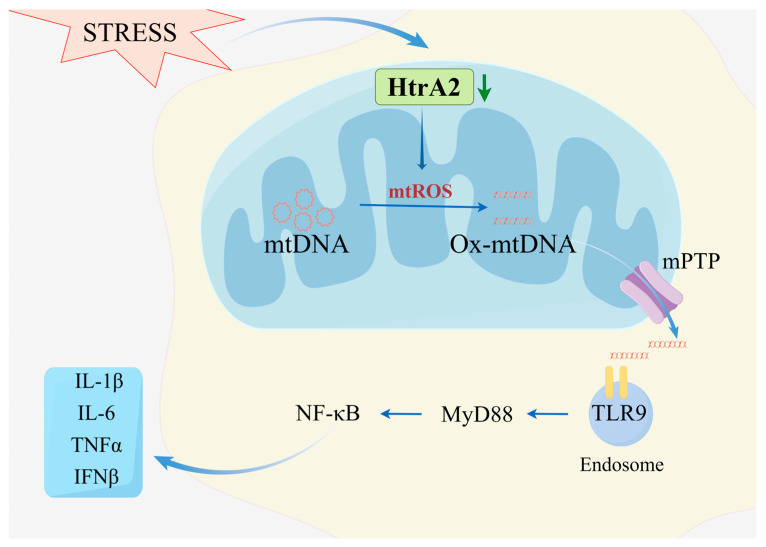
HtrA2/Omi deficiency is involved in pro-inflammatory macrophages formation by promoting Ox-mtDNA. Macrophages are chronically exposed to SCI and under stress conditions such as oxidative stress, heat stress, etc., mitochondrial membrane gap HtrA2/Omi is reduced. HtrA2/Omi deletion promotes an increase in the level of ROS leads to oxidative damage to mitochondrial mtDNA as well as induces the mitochondrial permeability transition pore MPTP to open, releasing Ox-mtDNA into the cytoplasm, which is then recognized by TLR9 (a receptor for DAMPs on macrophage endosomes), which activates innate immunity-related pathways to promote pro-inflammatory macrophages polarization and release inflammatory factors such as IL-1β, IL-6, TNFα, and IFNβ, which further exacerbate SCI progression. By Figdraw (https://www.figdraw.com, accessed on 1 December 2023.).

## Data Availability

Not applicable.

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
