# Peer review of "HTRA2/OMI-Mediated Mitochondrial Quality Control Alters Macrophage Polarization Affecting Systemic Chronic Inflammation"

_ijms, 2024, doi:10.3390/ijms25031577_

Round 1

Reviewer 1 Report

Comments and Suggestions for Authors

The article entitled “HTRA2/OMI-mediated mitochondrial quality control alters 2 macrophage polarization affecting Systemic chronic inflamma-3tion” is a well documented review on the literature on the role of HTRA2 in macrophage polarization and systemic chronic inflammation. I just have a few suggestions to improve the article.

Introduction, line 39-42. The nomenclature M1 and M2 is currently being questioned and some experts prefer to speak of pro-inflammatory and anti-inflammatory macrophages. Perhaps the authors could change their sentence.  

Line 49, the authors should clarify that IL-4 induces macrophages polarization into “anti-inflammatory macrophages”. 

Introduction: should the authors develop the abbreviation at the first use: SCI, FA, UPRmt, MQC.

A number of articles published in the 2000s demonstrated that cytoplasmic HtrA2 can bind and antagonize IAPs (Yang et al. Genes Dev; 2003, Hegde et al. J. Biol. Chem 2002, Varhagen et al. J. Biol. Chem 2002, Suzuki et al. Mol Cell 2001…). Since IAPs are potent regulators of inflammatory response and macrophage activity (Sharma et al. Immunol Cell Biol 2017, Moron-Calvente et al. PLoS Onse 2018, Nadella et al. Front Immunol 2018), I was wondering whether HtrA2 could affect systemic chronic inflammation through regulating proteins from IAP family? Do the authors think this point deserve to be discuss.   

Reviewer 2 Report

Comments and Suggestions for Authors

The review article by Liu et al provides an in-depth analysis of the involvement of HtrA2/Omi proteins in macrophage polarization and systemic chronic inflammation (SCI). The authors highlight that SCI, resulting from immune over-activation, is a key factor in the pathogenesis of various non-infectious chronic diseases, including neurodegenerative diseases and diabetes mellitus.

Dysregulation of macrophage polarization can contribute to the development and persistence of SCI. The review emphasizes the significance of HtrA2/Omi, an important regulator of mitochondrial quality control, in this context. The protein is involved not only in degradation of misaccumulated proteins through the mitochondrial unfolded protein response (UPRmt) but also in the regulation of mitochondrial dynamics and morphology. The paper summarizes the mechanisms by which HtrA2/Omi proteins modulate macrophage energy metabolism reprogramming, inflammatory signaling pathways, and mitochondrial quality control, ultimately impacting macrophage polarization remodeling.

Although the review is quite comprehensive, there are some details that the authors did not cover in the review, but they are relevant to the topic of the review.

1.      1. There are publications indicating that HtrA2/Omi is involved in the regulation of mitophagy, the selective autophagy of damaged mitochondria. The authors should pay more attention to this aspect.

2. HtrA2/Omi is involved in the regulation of mitochondrial biogenesis through interactions with the GSK3β/PGC-1α pathway.

3. The authors should give more attention in their review not only to mitochondrial dynamics, regulated by mitofusins, OPA1, and other proteins but also to the fundamental processes of mitochondrial pool renewal through the regulation of mitochondrial biogenesis and mitophagy by HtrA2/Omi.

In general, Liu et al offers a well-structured and informative review that contributes to our understanding of the role of HtrA2/Omi proteins in macrophage polarization and their implications in systemic chronic inflammation. The article will be of great interest to researchers and clinicians involved in the study and management of chronic inflammatory diseases.
